# Distinguishing between recruitment and spread of silent chromatin structures in *Saccharomyces cerevisiae*

Molly Brothers, Jasper Rine*

Department of Molecular and Cell Biology, University of California, Berkeley, Berkeley, United States

**Abstract** The formation of heterochromatin at *HML*, *HMR*, and telomeres in *Saccharomyces cerevisiae* involves two main steps: the recruitment of Sir proteins to silencers and their spread throughout the silenced domain. We developed a method to study these two processes at single basepair resolution. Using a fusion protein between the heterochromatin protein Sir3 and the nonsite-specific bacterial adenine methyltransferase M.EcoGII, we mapped sites of Sir3–chromatin interactions genome-wide using long-read Nanopore sequencing to detect adenines methylated by the fusion protein and by ChIP-seq to map the distribution of Sir3–M.EcoGII. A silencing-deficient mutant of Sir3 lacking its Bromo-Adjacent Homology (BAH) domain, *sir3-bahΔ*, was still recruited to *HML*, *HMR*, and telomeres. However, in the absence of the BAH domain, it was unable to spread away from those recruitment sites. Overexpression of Sir3 did not lead to further spreading at *HML*, *HMR*, and most telomeres. A few exceptional telomeres, like 6R, exhibited a small amount of Sir3 spreading, suggesting that boundaries at telomeres responded variably to Sir3-M.EcoGII overexpression. Finally, by using a temperature-sensitive allele of *SIR3* fused to *M.ECOGII*, we tracked the positions first methylated after induction and found that repression of genes at *HML* and *HMR* began before Sir3 occupied the entire locus.

*For correspondence:
jrine@berkeley.edu

Competing interest: The authors declare that no competing interests exist.

## Editor's evaluation

This manuscript studies the mechanism of transcriptional silencing in *S. cerevisiae*. Using two new tools for this field, fusion of a silencing protein to a DNA methyltransferase and long-read Nanopore sequencing, the results have provided both technical advances and new insights into the role of Sir3 in this process.

## Introduction

Cells have an interest in coordinating the expression of genes: It allows them to turn sets of genes on and off in response to various stimuli or ensure certain genes are always expressed or always repressed to create and maintain cell identity. There are multiple ways to coordinate transcription, including shared binding sites for activators or repressors in promoters, nuclear compartmentalization, and creation of large domains like heterochromatin. The establishment of coordinated, stable blocs of gene expression, such as heterochromatin, can be broken down into two main steps: nucleation, which involves the recruitment of chromatin-modifying factors, followed by the expansion of these chromatin-modifying factors beyond recruitment sites in an ill-defined process known as spreading.

An impressive example of the concepts of nucleation and spread is inactivation of the X chromosome in female mammals (reviewed in *Galupa and Heard, 2018*). Nucleation begins with the transcription of the noncoding RNA *Xist* from one of the two X chromosomes, which recruits heterochromatin

factors in cis that eventually coat and transcriptionally silence nearly the entire 167-megabase X chromosome. Two studies have characterized early steps in nucleation at the *Xist* locus and at recruitment sites throughout the X chromosome (*Engreitz et al., 2013*; *Simon et al., 2013*), but the mechanics of how the XIST transcript and associated heterochromatin proteins spread remains unclear even for this well-studied phenomenon. Furthermore, 'spreading' itself is an inferred process that connects known recruitment sites to the final binding profile of heterochromatin proteins. A clear mechanistic distinction between nucleation and spread, with a characterization of intermediate steps, has not been achieved for any organism.

Transcriptional silencing in the budding yeast *Saccharomyces cerevisiae* is one of the best-studied heterochromatic phenomena (reviewed in *Gartenberg and Smith, 2016*). Heterochromatin is created by the S̲ilent I̲nformation R̲egulator (SIR) complex that silences the transcription of genes at *HML* and *HMR* and the 32 telomeres. The recruitment of the SIR complex to *HML*, *HMR*, and telomeres is sequence specific, whereas its spreading is sequence nonspecific. More specifically, different combinations of Rap1, Abf1, and Origin Recognition Complex (ORC) binding sites are present at the *E* and *I* silencers that flank *HML* and *HMR* (*Buchman et al., 1988*; *Kimmerly et al., 1988*; *Shore et al., 1987*; *Shore and Nasmyth, 1987*) and at the TG repeats and X elements of telomeres (*Buchman et al., 1988*; *Longtine et al., 1989*; *Stavenhagen and Zakian, 1994*). These proteins in turn interact with and recruit the SIR complex (*Cockell et al., 1995*; *Moretti and Shore, 2001*; *Triolo and Sternglanz, 1996*). The SIR complex then deacetylates chromatin (*Braunstein et al., 1993*; *Ellahi et al., 2015*; *Suka et al., 2001*; *Thurtle and Rine, 2014*), resulting in chromatin compaction (*Georgel et al., 2001*; *Gottschling, 1992*; *Johnson et al., 2009*; *Loo and Rine, 1994*; *Singh and Klar, 1992*; *Swygert et al., 2018*). As a result, transcription is blocked at least in part by steric occlusion, though details remain unknown (*Chen and Widom, 2005*; *Gao and Gross, 2008*; *Johnson et al., 2013*; *Lynch and Rusche, 2009*; *Sekinger and Gross, 2001*; *Steakley and Rine, 2015*). Almost any gene placed within the defined domain can be transcriptionally silenced, establishing the locus-specific, gene nonspecific nature of heterochromatic silencing (*Dodson and Rine, 2015*; *Gottschling et al., 1990*; *Saxton and Rine, 2019*; *Schnell and Rine, 1986*; *Sussel et al., 1993*). This difference in sequence dependence between recruitment and spread implies they are separable processes that rely on different factors and interactions.

Of the three SIR complex members (Sir2, Sir3, and Sir4), Sir3 is thought to be the major structural driver of heterochromatin spread and compaction. Sir3 interacts with the silencer-binding proteins Abf1 and Rap1 (*Moretti et al., 1994*; *Moretti and Shore, 2001*), with the other members of the SIR complex, Sir2 and Sir4 (*Chang et al., 2003*; *Ehrentraut et al., 2011*; *Rudner et al., 2005*; *Samel et al., 2017*; *Strahl-Bolsinger et al., 1997*), with nucleosomes (*Armache et al., 2011*; *Hecht et al., 1995*; *Johnson et al., 1990*; *Norris et al., 2008*; *Onishi et al., 2007*; *Wang et al., 2013*), and with itself (*King et al., 2006*; *Liaw and Lustig, 2006*; *Oppikofer et al., 2013*). All of these Sir3 interactions are required for transcriptional silencing. In vitro, Sir3 dimers can bridge neighboring nucleosomes and compact chromatin (*Behrouzi et al., 2016*). Among the SIR complex members, Sir3 has the largest difference in affinity for acetylated and deacetylated histone tails (*Armache et al., 2011*; *Carmen et al., 2002*; *Onishi et al., 2007*; *Oppikofer et al., 2011*; *Swygert et al., 2018*). By binding deacetylated histone tails more strongly than acetylated ones, Sir3 helps create a positive feedback loop wherein Sir2 deacetylates histone tails (*Ghidelli et al., 2001*; *Imai et al., 2000*; *Landry et al., 2000*; *Smith et al., 2000*) and Sir3 reinforces and further recruits Sir2/4 to silent regions.

Characterization of SIR complex nucleation and spread is limited by techniques like ChIP-seq that measure processes on populations of molecules and at a resolution limited by sequencing-read length. We developed a new method that allowed us to characterize the binding of heterochromatin proteins at basepair resolution. Using long-read sequencing, we used this new method to resolve the distinction between the recruitment and spread of Sir3 and to track the establishment of gene silencing in heterochromatin over time. These data pinpointed when the process of transcriptional silencing begins.

## Results

### The Sir3–M.EcoGII fusion protein strongly and specifically methylated *HML* and *HMR*

To achieve a higher-resolution method for assessing SIR complex binding, we made a fusion protein between Sir3 and M.EcoGII (*Figure 1A*), a nonsite-specific bacterial N6-methyladenosine methyltransferase (*Murray et al., 2018*; *Woodcock et al., 2020*). In principle, wherever Sir3 binds chromatin, even transiently, M.EcoGII has the opportunity to methylate nearby accessible adenines to make m⁶A. *S. cerevisiae* has no endogenous DNA methylation and no demethylases, allowing us to attribute m⁶A only to the activity of the fusion protein, reflecting where it resides as well as where it has been. M.EcoGII has no specific recognition sequence, which should provide more resolution than methyltransferases like *E. coli* Dam, which has a four basepair recognition site. The positions of Sir3–M.EcoGII can be determined conventionally by immunoprecipitation with an antibody against m⁶A followed by sequencing the precipitated DNA using Illumina sequencing. The more powerful implementation would come from distinguishing between individual m⁶A bases and unmodified A bases using long-read Nanopore sequencing (*Xu and Seki, 2020*).

To test this concept, we first assessed the silencing ability of Sir3–M.EcoGII by measuring mRNA expression of *HMLα2* and *HMRa1*. Strains expressing Sir3–M.EcoGII could silence *HML* and *HMR* as well as wild-type strains compared to *sir3Δ* strains that displayed a full loss of silencing (*Figure 1B*). Thus, the fusion protein retained full function of Sir3.

To compare the binding profile of Sir3–M.EcoGII to the distribution of methylation it produced, we performed DNA immunoprecipitation and Illumina sequencing (DIP-seq) using an antibody that specifically recognizes m⁶A alongside ChIP-seq for a V5 epitope-tagged Sir3–M.EcoGII. ChIP-seq revealed strong Sir3–M.EcoGII occupancy over the *E* and *I* silencers at *HML* and over the *E* silencer at *HMR* with weaker but consistent signal above background between the two silencers (*Figure 1C*, top row). Compared to ChIP-seq, methylation measured by DIP-seq had a stronger signal over the entirety of *HML* and *HMR* and a broader signal that extended beyond the silencers (*Figure 1C*, *Figure 1—figure supplement 1*). The methylation over *HML* and *HMR* was from the fusion protein Sir3–M.EcoGII, as neither a strain without M.EcoGII nor a strain expressing unfused M.EcoGII from the SIR3 promoter showed appreciable DIP-seq signal (*Figure 1C*, *Figure 1—figure supplement 1*).

Methylation by Sir3–M.EcoGII measured by Nanopore sequencing agreed well with DIP-seq, showing strong methylation over *HML* and *HMR* with little background methylation outside of these regions (*Figure 1D*, *Figure 1—figure supplement 2A*). Fusions of Sir2 and Sir4 with M.EcoGII were also fully silencing competent (*Figure 1B*) and produced methylation signals that matched Sir3–M.EcoGII at *HML* and *HMR* (*Figure 1D*), suggesting that all three members of the SIR complex were equally distributed, as expected. In addition to the aggregate methylation signal (% of reads methylated at each position), we analyzed methylation of single adenines on single reads across *HML* and *HMR* (*Figure 1E*, *Figure 1—figure supplement 2B*). Sir3–M.EcoGII methylated most strongly near the *E* and *I* silencers and at the *HMLα1/α2* promoter, with lower, but significant, methylation between these sites (*Figure 1E*, *Figure 1—figure supplement 2B*). Analysis of single reads revealed a periodicity of methylation across *HML* and *HMR* (*Figure 1E*, *Figure 1—figure supplement 2B*). These small regions of higher methylation corresponded to linker regions between nucleosomes (*Figure 1—figure supplement 3*). Sir3–M.EcoGII did methylate within nucleosome-occupied regions at *HML* and *HMR* but at a lower frequency (*Figure 1—figure supplement 3*), consistent with in vitro studies that use methylation by M.EcoGII or another nonspecific adenine methyltransferase, Hia5, as a measurement of chromatin accessibility (*Abdulhay et al., 2020*; *Brady et al., 2021*; *Shipony et al., 2020*; *Stergachis et al., 2020*).

### Nucleosome binding was required for spreading, but not recruitment, of Sir3

The recruitment of the SIR complex to silencers is sequence specific. Rap1, Abf1, and ORC bind at these recruitment sites and recruit the Sir proteins directly through protein–protein interactions. In contrast, SIR complex binding away from recruitment sites is sequence independent and instead relies on interactions with nucleosomes and among the Sir proteins themselves.

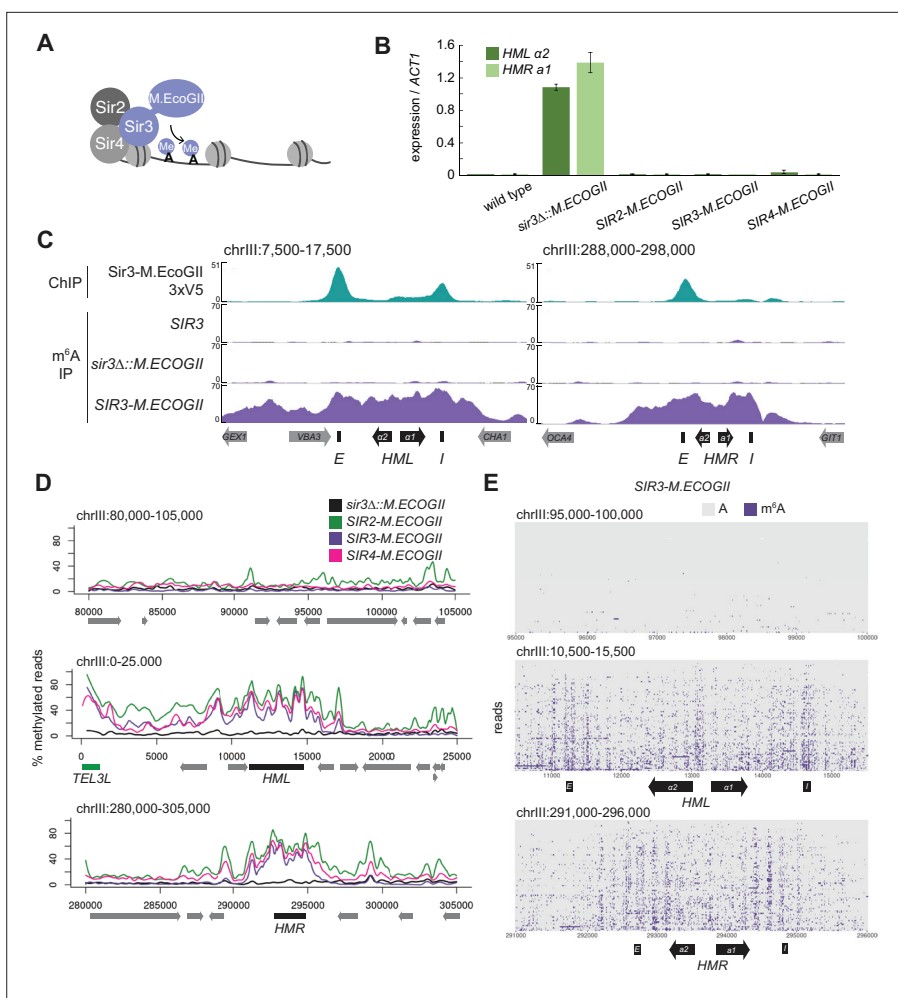

**Figure 1.** The Sir3–M.EcoGII fusion protein strongly and specifically methylated *HML* and *HMR*. (**A**) Sir3–M.EcoGII is a fusion protein that nonspecifically methylates adenines in regions that Sir3 binds. (**B**) RT-qPCR of *HMLα2* and *HMRa1* mRNA, normalized to *ACT1* mRNA, in strains expressing no fusion proteins (wild type, JRY11699, JRY9316), *sir3Δ::M.ECOGII* (JRY13029, JRY13030), *SIR2-M.ECOGII* (JRY13625, JRY13626), *SIR3-M.ECOGII* (JRY12840, JRY13027), and *SIR4-M.ECOGII* (JRY13021, JRY13022). Data are the average of three biological replicates, and bars mark one standard deviation. (**C**) ChIP-seq of Sir3-M.EcoGII-3xV5 (top row, JRY12839) and DNA m6A immunoprecipitation and sequencing (DIP-seq) of no EcoGII (row two, JRY11699), *sir3Δ::M.ECOGII* (row three, JRY12838), and *SIR3-M.ECOGII* (row four, JRY12840). Shown are 10 kb regions centered at *HML* (left) and *HMR* (right). Input results are plotted but not visible due to the strong ChIP-seq and DIP-seq signals. (**D**) Aggregate results from long-read Nanopore sequencing of *sir3Δ::M.ECOGII* (black line, JRY12838), *SIR2-M.ECOGII* (green line, JRY13625), *SIR3-M.ECOGII* (purple line, JRY13027), and *SIR4-M.ECOGII* (pink line, JRY13021). The *y*-axis represents the percentage of reads in each position called as methylated by the modified base-calling software Megalodon (see Materials and methods). Shown are 25 kb windows at a control region on chromosome III to show background methylation (top row), at *HML* (middle row), and at *HMR* (bottom row). (**E**) Single-read plots from long-read Nanopore sequencing of *SIR3-M.ECOGII* (JRY13027). Each row of the plots is a single read the spans the entire query region, ordered by lowest average methylation on the top to highest average methylation on the bottom. Methylated adenines are colored purple, and unmethylated adenines are colored gray. Shown are 5 kb windows at a control region on chromosome III to show background methylation (top row), at *HML* (middle row), and at *HMR* (bottom row). M.EcoGII nor a strain expressing unfused M.EcoGII from the *SIR3* promoter showed appreciable DIP-seq signal (**C**, *Figure 1—figure supplement 1*).

The online version of this article includes the following figure supplement(s) for figure 1:

**Figure supplement 1.** *Sir3–M.EcoGII* strongly and specifically methylated *HML* and *HMR*.

**Figure supplement 2.** *Sir3–M. EcoGII* strongly and specifically methylated *HML* and *HMR*.

**Figure supplement 3.** Sir3–M.EcoGII preferentially methylated linker regions.

We therefore hypothesized that the interaction between Sir3 and nucleosomes would not be required for nucleation at recruitment sites but would be required for binding outside of those recruitment sites. Sir3 has multiple recognized domains (*Figure 2A*): The Bromo-Adjacent Homology (BAH) domain, which interacts with nucleosomes (*Armache et al., 2011*; *Buchberger et al., 2008*; *Norris et al., 2008*; *Onishi et al., 2007*; *Rudner et al., 2005*; *Sampath et al., 2009*), the AAA+ domain, which interacts with Sir4 (*Ehrentraut et al., 2011*; *King et al., 2006*; *Samel et al., 2017*), and the winged helix (wH) domain, which allows for homodimerization (*King et al., 2006*; *Liaw and Lustig, 2006*; *Oppikofer et al., 2013*). We deleted the BAH domain of Sir3–M.EcoGII (*bahΔ*), which largely abrogates Sir3–nucleosome interactions in vitro (*Buchberger et al., 2008*; *Onishi et al., 2007*; *Sampath et al., 2009*). Previous studies established that deletion of the BAH domain causes a phenotypic loss of silencing, likely due to a loss of interaction with nucleosomes (*Buchberger et al., 2008*; *Gotta et al., 1998*; *Onishi et al., 2007*), but did not characterize its binding at regions of heterochromatin. Importantly for what follows, deletion of the BAH domain did not destabilize Sir3 (*Figure 2B*). We also confirmed that *sir3-bahΔ-M.ECOGII* strains displayed a loss of silencing, but found that the loss was more severe at *HML* than at *HMR* (*Figure 2C*).

Despite the loss of silencing at *HML* and *HMR* in the *sir3-bahΔ-M.ECOGII* strain, there was still detectable methylation across the two loci (*Figure 2D, E*, *Figure 2—figure supplement 1A*). At the aggregate level, sir3-bahΔ-M.EcoGII methylated silencers at *HML* and *HMR* at the same level as Sir3–M.EcoGII but showed decreased methylation between them (*Figure 2D*). Analysis of single reads spanning *HML* and *HMR* in the *bahΔ* mutant showed similarly strong levels of methylation at silencers, and revealed strong methylation both at the promoters of *HML* and *HMR* and the recognition site for the HO endonuclease (*Figure 2E*). In contrast, little methylation was seen over gene bodies between these sites (*Figure 2E*).

The ability of Nanopore to sequence long reads also allowed mapping and analysis of Sir3 occupancy on the repetitive and highly homologous telomeres, an issue with short-read Illumina sequencing. In addition to methylating *HML* and *HMR* (*Figure 1*), Sir3–M.EcoGII strongly methylated telomeres at TG repeats and X elements (*Figure 2F*, *Figure 2—figure supplement 1B*, *Figure 2—figure supplement 2*), and the periodicity of methylation was apparent on single reads as well (*Figure 2G*), likely corresponding to more-accessible linker regions. The loss of binding outside of recruitment sites of sir3-bahΔ-M.EcoGII was more striking at telomeres than *HML* and *HMR*, where methylation by the *bahΔ* mutant matched wild-type levels at TG and X repeats but dropped off steeply centromere-proximal to the X elements (*Figure 2F, G*, *Figure 2—figure supplement 1B*, *Figure 2—figure supplement 2*). The results at telomeres, supported by the data at *HML* and *HMR*, suggested that the nucleosome-binding activity of Sir3 was required for Sir3 to spread away from recruitment sites, but not for its initial recruitment.

## *SIR3* expression level did not limit its spread from recruitment sites

In addition to understanding what enables Sir3 spreading, we were also interested in what limits its spread. One common feature of heterochromatin proteins is that their activity is dose dependent: lowered expression causes loss of heterochromatin whereas elevated expression can cause silencing of genes near heterochromatin (*Henikoff, 1996*; *Locke et al., 1988*). Indeed, it was previously reported that overexpression of Sir3 results in its spread beyond wild-type boundaries, accompanied by repression of genes in those extended regions (*Hecht et al., 1996*; *Hocher et al., 2018*; *Ng et al., 2003*; *Renauld et al., 1993*; *Strahl-Bolsinger et al., 1997*). To provide an independent test of those conclusions, we tested whether the expression level of Sir3 limited how far it could spread beyond recruitment sites at *HML*, *HMR*, and telomeres. mRNA levels of *SIR3-M.ECOGII* on a multicopy 2μ plasmid was tenfold higher than from the chromosomal *SIR3-M.ECOGII* locus in a strain carrying an empty 2μ plasmid (*Figure 3A*). We confirmed the overexpression of *SIR3-M.ECOGII* in these strains by protein immunoblotting (*Figure 3B*). Overexpression of *SIR3-M.ECOGII* had no effect on silencing at *HML* and *HMR* (*Figure 3C*).

Overexpression of *SIR3-M.ECOGII* had little effect on the boundaries of methylation at *HML* and *HMR*. Strains overexpressing *SIR3-M.ECOGII* displayed increased methylation over both loci and between *HML* and telomere 3 L, but no new sites of methylation appeared outside the bounds of strains expressing only one copy of *SIR3-M.ECOGII* (*Figure 3D*). The changes in methylation upon overexpression of Sir3–M.EcoGII matched changes in occupancy of the fusion protein by ChIP-seq,

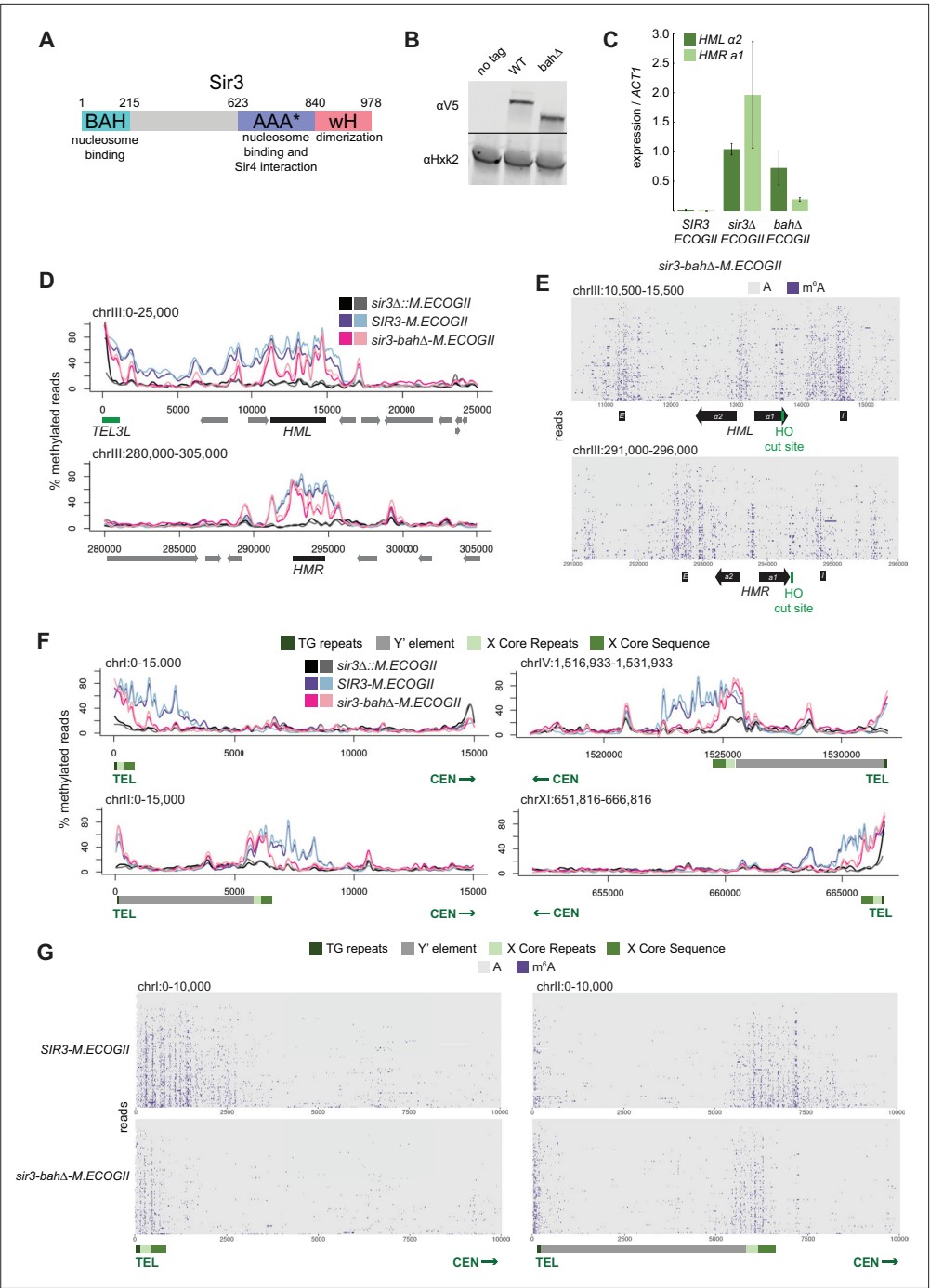

**Figure 2.** Nucleosome binding was required for spread, but not recruitment, of Sir3 to regions of heterochromatin. (**A**) Schematic of Sir3 protein domains. (**B**) Protein immunoblotting in strains expressing Sir3 (no tag, JRY11699), Sir3-3xV5 (JRY12601), and sir3-bahΔ–3xV5 (JRY13621). Top row are 3xV5-tagged Sir3 proteins, and bottom row is the loading control Hxk2. The unedited blot is in **Figure 2—source data 1**. (**C**) RT-qPCR of *HMLα2* and *HMRa1* mRNA, normalized to *ACT1* mRNA, in strains expressing *SIR3-M.ECOGII* (JRY12840, JRY13027), *sir3Δ::M.ECOGII* (JRY13029, JRY13030), and *sir3-bahΔ-M.ECOGII* (JRY13438, JRY13439). Data are the average of three biological replicates, and bars mark one standard deviation. (**D**) Aggregate methylation results at *HML* (top) and *HMR* (bottom) from long-read Nanopore sequencing of *sir3Δ::M.ECOGII* (JRY13029, JRY13030), *SIR3-M.ECOGII* (JRY12840, JRY13027), and *sir3-bahΔ-M.ECOGII* (JRY13438). Plots are as described in **Figure 1D**. (**E**) Single-read plots from long-read Nanopore sequencing of *sir3-bahΔ-M.ECOGII* (JRY13438) at *HML* (top) and *HMR* (bottom). Plots are as described in **Figure 1E**. (**F**) Aggregate methylation results at four representative telomeres (1 L, 2 L,

*Figure 2 continued on next page*

*Figure 2 continued*

4 R, and 11 R) from long-read Nanopore sequencing of the same strains as **D**. Shown are 15 kb windows of each telomere. Plots are as described in *Figure 1D*. (**G**) Single-read plots from long-read Nanopore sequencing of *SIR3-M.ECOGII* (JRY13027) and *sir3-bahΔ-M.ECOGII* (JRY13438) at two representative telomeres (1 L and 2 L). Shown are 10 kb windows of each telomere.

The online version of this article includes the following source data and figure supplement(s) for figure 2:

**Source data 1.** Uncropped protein immunoblot of Sir3 mutants.

**Figure supplement 1.** DIP-seq of *SIR3-M.ECOGII* (top row, JRY13027), *sir3Δ::M.ECOGII* (middle row, JRY13030), and *sir3-bahΔ-M.ECOGII* (bottom row, JRY13438).

**Figure supplement 2.** Methylation by Sir3–M.EcoGII and sir3-bahΔ-M.EcoGII at all 32 telomeres.

suggesting that methylation reflects binding of the protein well (*Figure 3—figure supplement 1*). There was some increase in methylation over the promoters of two genes closest to *HML* and *HMR* (*CHA1* and *OCA4*, respectively), but it did not result in any change in the level of their expression (*Figure 3E*).

Surprisingly, the results at telomeres were qualitatively similar but revealed three categories of effects. Some telomeres showed a large increase in the amount of methylation upon overexpression of *SIR3-M.ECOGII* with a small extension of binding farther into the chromosome (*Figure 3F*, top row, *Figure 3—figure supplement 2*). Some telomeres showed a modest increase in the amount methylation upon overexpression of *SIR3-M.ECOGII* with little, if any, extension of range (*Figure 3F*, middle row, *Figure 3—figure supplement 2*). Finally, some telomeres showed no appreciable change in methylation (*Figure 3F*, bottom row, *Figure 3—figure supplement 2*). As at *HML* and *HMR*, the changes in methylation seen with Nanopore sequencing matched with changes in occupancy measured by ChIP-seq (*Figure 3—figure supplement 1*). Only three of the 32 telomeres, including the paradigmatic telomere 6 R from earlier studies (*Hecht et al., 1996*; *Renauld et al., 1993*; *Strahl-Bolsinger et al., 1997*), showed convincing spread of methylation to new sites compared to telomeres in strains expressing one copy of *SIR3-M.ECOGII* (*Figure 3F*, *Figure 3—figure supplement 2*). Therefore, the expression level of *SIR3* was not a universal limiting factor in heterochromatin spread. These results imply the existence of other chromatin features that create boundaries for Sir3 spreading.

## Repression of *HML* and *HMR* preceded heterochromatin maturation

To evaluate the dynamics of Sir3 recruitment and spreading during the establishment of heterochromatin over time, we used a temperature-sensitive allele of *SIR3*, *sir3-8*, fused to *M.ECOGII* and took samples at various time points for Nanopore sequencing after switching from the restrictive (37°C) to the permissive (25°C) temperature (*Figure 4A*). In agreement with previous studies (*Stone et al., 2000*), growth at 37°C caused lower protein levels of *sir3-8* (*Figure 4B*). Over the course of 150 min, the protein levels of *sir3-8* slowly increased to 63% of the level in constitutive 25°C growth conditions (*Figure 4B*). In cells grown at 37°C, *sir3-8-M.ECOGII* did not methylate *HML* and *HMR*, but when grown constitutively at 25°C, there was strong methylation over both loci (*Figure 4C*, *Figure 4—figure supplement 1*).

Over a 90-min time course, methylation increased only over the silencers and promoters of *HML* and *HMR* (*Figure 4D*, *Figure 4—figure supplement 2*, solid lines). Methylation at the promoter of *HML* during the time course and in the *sir3-bahΔ* mutant was expected, as it contains a Rap1-binding site, and Rap1 interacts directly with Sir3 and Sir4. However, methylation at the promoter of *HMR* at these early time points and in the *sir3-bahΔ* mutant was a surprise, as there is not a known SIR complex-interacting protein that binds at the promoter. In the absence of Sir3, Sir4 is still bound at silencers (*Goodnight and Rine, 2020*; *Hoppe et al., 2002*; *Rusché et al., 2002*), so the faster recruitment at these sites, and perhaps the promoters as well, might be due to the interaction of Sir3 with Sir4 and Rap1.

By 90 min (~1 cell division), methylation at no position reached the level found in cells constitutively grown at 25°C – the level of methylation of mature, stable heterochromatin (*Figure 4D*, *Figure 4—figure supplement 2*, dotted line). Strikingly, even by 30 min after the temperature shift, when methylation was just rising above background at silencers, partial repression of *HML* and *HMR*

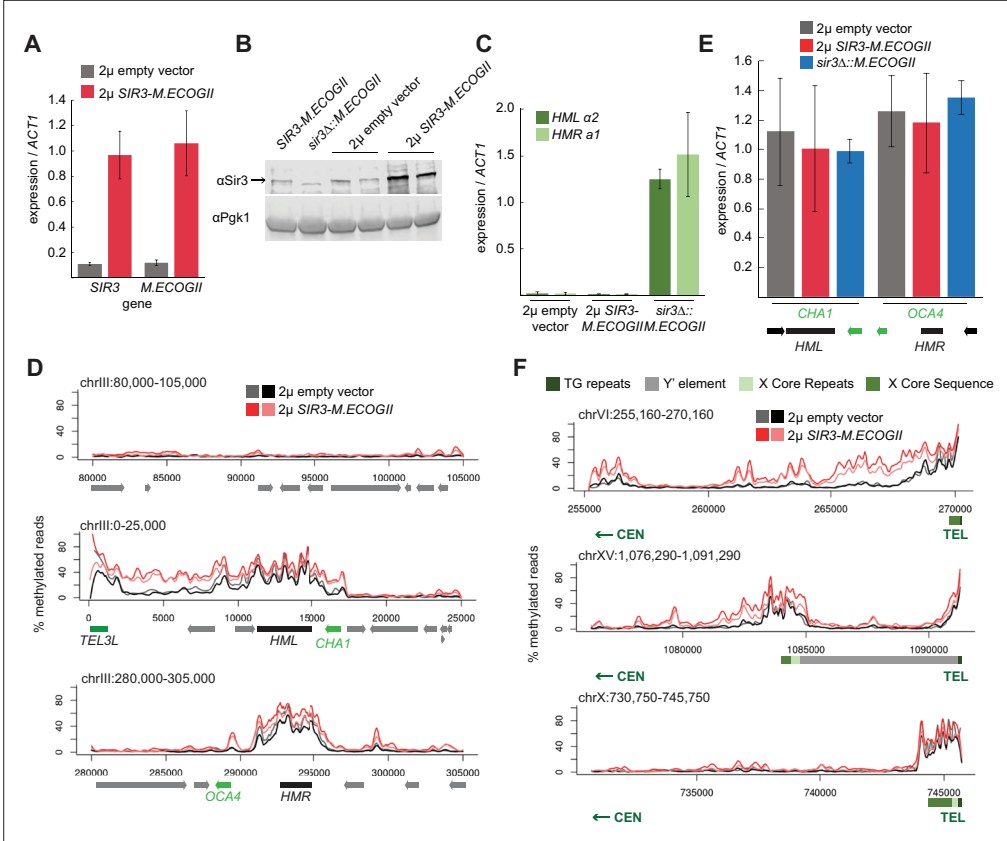

**Figure 3.** Sir3 expression level was not limiting for its spread from recruitment sites. (**A**) RT-qPCR of *SIR3* and *M.ECOGII* mRNA normalized to *ACT1* mRNA in *SIR3-M.ECOGII* strains carrying an empty multicopy 2μ vector (JRY13670, JRY13671) and strains carrying a multicopy 2μ *SIR3-M.ECOGII* plasmid (JRY13672, JRY13673). Data are the average of four biological replicates, and bars mark one standard deviation. (**B**) Protein immunoblotting in the same strains as **A** (JRY13670-JRY13671) as well as *SIR3-M.ECOGII* (no plasmid, JRY13027) and *sir3Δ::M.ECOGII* (JRY13029). Top row is Sir3, and bottom row is the loading control Pgk1. The unedited blot is in ***Figure 3—source data 1***. (**C**) RT-qPCR of *HMLα2* and *HMRa1* mRNA normalized to *ACT1* mRNA in the same strains as **A** as well as *sir3Δ::M.ECOGII* (JRY13029, JRY13030). Data are the average of four biological replicates, and bars mark one standard deviation. (**D**) Aggregate methylation results at a control region on chromosome III to show background levels of methylation (top row), at *HML* (middle row) and *HMR* (bottom row) from long-read Nanopore sequencing of the same strains in **A**. The two colors for each genotype correspond to two biological replicates. Plots are as described in **D**. (**E**) RT-qPCR of CHA1 and OCA4 mRNA normalized to ACT1 mRNA in the same strains as **C**. (**F**) Aggregate methylation results at three representative telomeres (6 R, 15 R, and 10 R) from long-read Nanopore sequencing of the same strains as **A**. The two colors for each genotype correspond to two biological replicates. Shown are 15 kb windows of each telomere. Plots are as described in ***Figure 1D***.

The online version of this article includes the following source data and figure supplement(s) for figure 3:

**Source data 1.** Uncropped Protein Immunoblot of Sir3-M.EcoGII overexpression strains.

**Figure supplement 1.** Binding of Sir3–M.EcoGII measured by ChIP-seq upon overexpression of *SIR3-M.ECOGII*.

**Figure supplement 2.** Methylation upon overexpression of *SIR3-M.ECOGII* at all 32 telomeres.

---

was apparent (***Figure 4E***). This result suggested that binding of Sir3 at silencers and promoters was sufficient for partial repression and preceded its spread over the entirety of both loci.

## Discussion

In this study, we developed a new method to study the process of recruitment and spread of the *S. cerevisiae* heterochromatin protein Sir3 in living cells with a resolution approximating the frequency of single A–T basepairs. We created a fusion protein between a key structural protein of heterochromatin, Sir3, and the bacterial adenine methyltransferase M.EcoGII that retained function and activity of

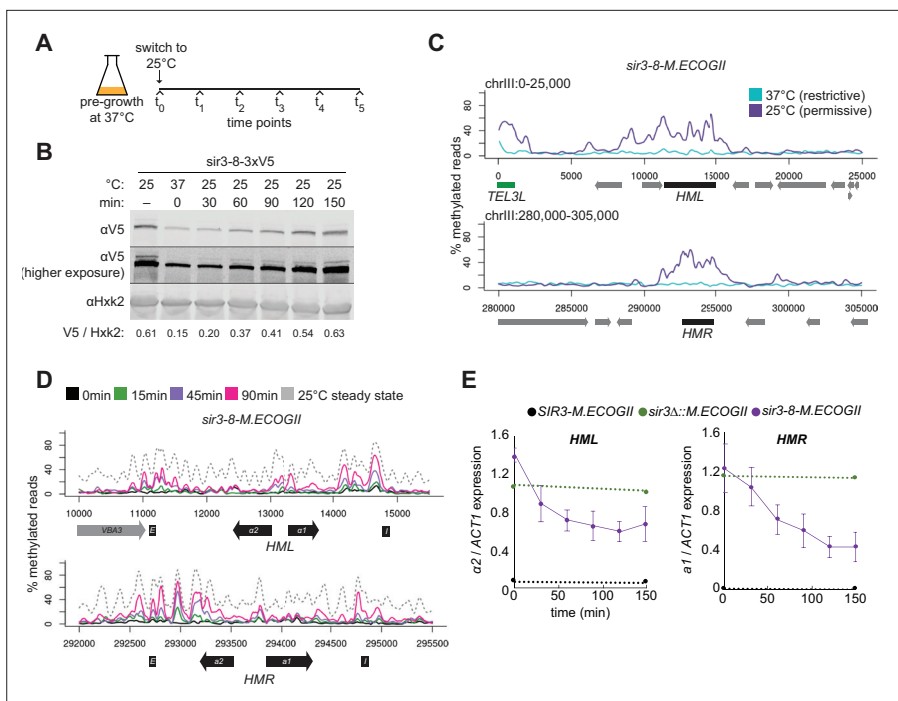

**Figure 4.** Repression of *HML* and *HMR* preceded heterochromatin maturation. (**A**) Schematic of temperature-shift time course with *sir3-8-M.ECOGII*. (**B**) Protein immunoblotting in a strain expressing sir3-8-3xV5 (JRY13467) constitutively at 25°C (first lane), constitutively at 37°C (second lane), and at 30, 60, 90, 120, and 150 min after a shift to 25°C. Top row is 3xV5-tagged sir3-8 protein, the middle row is the same as the top row but at a higher exposure, and the bottom row is the loading control Hxk2. The unedited blot is in . (**C**) Aggregate methylation results at *HML* (top) and *HMR* (bottom) from long-read Nanopore sequencing of strains expressing *sir3-8-M.ECOGII* (JRY13114) grown constitutively at 25 or 37°C. Plots are as described in *Figure 1D*. (**D**) Aggregate methylation results at *HML* (top) and *HMR* (bottom) from long-read Nanopore sequencing of a strain expressing *sir3-8-M.ECOGII* (JRY13134) grown constitutively at 25°C (dotted gray line) and collected at 0, 15, 45, and 90 min after a temperature switch from 37 to 25°C. (**E**) RT-qPCR of *HMLα2* (left) and *HMRa1* (right) mRNA in strains expressing *SIR3-M.ECOGII* (black, JRY13027, JRY12840), *sir3Δ::M.ECOGII* (green, JRY13029, JRY13030), or *sir3-8-M.ECOGII* (purple, JRY13114, JRY13134) collected at 0, 30, 60, 90, 120, and 150 min after a temperature switch from 37 to 25°C. Points are the average of three biological replicates and bars mark one standard deviation.

The online version of this article includes the following figure supplement(s) for figure 4:

**Figure supplement 1.** DIP-seq of *sir3-8-M.ECOGII* (JRY13114).

**Figure supplement 2.** Nanopore sequencing over temperature switch time course (biological replicate).

each. We used DNA methylation as a read-out for Sir3 occupancy on chromatin. Long-read Nanopore sequencing allowed us to distinguish directly between methylated and unmethylated adenine and transcended the limitations of earlier studies imposed by repetitive regions common at telomeres.

The methylation by Sir3–M.EcoGII at *HML* and *HMR* was stronger and had a larger footprint than its occupancy as judged by ChIP-seq, suggesting that our method captured transient contacts of Sir3 with chromatin that ChIP-seq could not. This result reinforced the idea that protein–chromatin interactions are dynamic, even for a feature like heterochromatin, commonly thought of as 'stable'. We also harnessed the power of single basepair resolution afforded by Nanopore sequencing to distinguish between recruitment and spread of Sir3 by studying a mutant of Sir3 whose distribution and binding profile had not yet been characterized, *sir3-bahΔ*. This mutant cannot bind to nucleosomes and loses transcriptional silencing at *HML* and *HMR* (**Buchberger et al., 2008**; **Gotta et al., 1998**; **Onishi et al., 2007**). The mutant sir3-bahΔ-M.EcoGII protein was still recruited to silencers at *HML*, *HMR*, and to telomeres where it methylated local adenines. However, the mutant did not spread beyond those recruitment sites, unlike wild-type Sir3–M.EcoGII. Our findings supported that recruitment and spread were separate processes that involved different interactions between Sir3 and other proteins at silenced loci and telomeres. The separation of nucleation and spread

mechanisms are also seen in studies of *S. pombe* heterochromatin. In *S. pombe*, transcriptional silencing of mating-type loci involves recruitment of histone deacetylases, histone methyltransferases, and the chromatin-binding factor Swi6/HP1 to sequence-specific nucleation sites followed by spreading of these factors over mating-type gene bodies to silence them. Most notably, spreading, but not nucleation, requires a 'read-write' mechanism in which the complexes that methylate histone H3 on lysine 9 (H3K9) also recognize that methylation mark (*Akoury et al., 2019*; *Al-Sady et al., 2013*; *Ivanova et al., 1998*; *Zhang et al., 2008*). Without this 'read-write' mechanism in place, heterochromatin factors are recruited, but they do not spread beyond their nucleation sites (*Noma et al., 2004*; *Yamada et al., 2005*; *Zhang et al., 2008*).

The ability to unambiguously map long reads to telomeres allowed us to challenge historic conclusions about Sir3 dosage-driven heterochromatin spreading. Overexpressing Sir3–M.EcoGII increased the methylation signal where there already was methylation at endogenous levels of expression, including a large increase in methylation between *HML* and telomere 3 L. However, the boundaries where methylation dropped to *sir3Δ::M.ECOGII* levels at *HML*, *HMR*, and telomeres mainly remained fixed in the two conditions. This result suggested that overexpression of Sir3 was not sufficient for spreading past most wild-type boundaries. The original idea that overexpression of Sir3 results in its further spread relied on low-resolution RT-PCR at two telomeres, 5 R and 6 R (*Hecht et al., 1996*; *Renauld et al., 1993*; *Strahl-Bolsinger et al., 1997*). In our genome-wide analysis, telomere 6 R was an exception, not the rule, as most telomeres did not show spreading of Sir3 upon its overexpression. We were not able to reproduce the result at telomere 5 R (*Hecht et al., 1996*).

Our data suggested that binding of Sir3 outside of *HML*, *HMR*, X elements, and telomere TG repeats was probabilistic: When Sir3 was overexpressed, its interactions outside of recruitment sites became more frequent, thus increasing the methylation signal produced by Sir3–M.EcoGII. However, most boundaries were left largely unchanged under overexpression conditions, in agreement with more recent Sir3 ChIP-seq results (*Radman-Livaja et al., 2011*). Perhaps other features, such as transcription, tRNA genes (*Donze and Kamakaka, 2001*; *Simms et al., 2008*; *Valenzuela et al., 2009*), the presence of histone variants like H2A.Z (*Babiarz et al., 2006*; *Giaimo et al., 2019*; *Meneghini et al., 2003*; *Venkatasubrahmanyam et al., 2007*), or the presence of Sir3-inhibiting chromatin marks like methylation of H3 on lysine 79 (*Altaf et al., 2007*; *Ng et al., 2003*; *Ng et al., 2002*; *Oki et al., 2004*; *Park et al., 2002*; *Stulemeijer et al., 2011*), enforce a boundary that overexpression of Sir3 itself cannot overcome.

Two results suggested that repression of *HML* and *HMR* did not require that Sir3 occupy the entire locus to the level seen in wild-type cells. During the temperature switch time course, methylation by sir3-8-M.EcoGII appeared first and most strongly at promoters and silencers of *HML* and *HMR*, with little or no detectable methylation between these sites. Yet, partial transcriptional repression was already evident at both loci within 30 min. The gradual repression that appeared during the time course was consistent with single-cell studies that show gradual tuning down of transcription during silencing establishment (*Goodnight and Rine, 2020*). The level of repression achieved during the time course was commensurate with that expected within the first cell cycle following restoration of Sir3 function (*Goodnight and Rine, 2020*). We also found that *sir3-bahΔ*, a nucleosome-binding mutant that was recruited to silencers and promoters but could not bind outside of them, achieved some repression at both *HML* and *HMR*. Repression was stronger at *HMR* than at *HML* in the *bahΔ* mutant, perhaps because *HMR* is smaller and less dependent on spreading. The time course and *bahΔ* results together suggested a difference between repression of transcription per se and the stability of silencing ultimately achieved by the SIR complex binding over the entirety of *HML* and *HMR*.

We focused our efforts on the heterochromatin protein Sir3 in *S. cerevisiae*, but this method may have broad utility. By relaxing the requirement for stable binding to detect interaction of a protein with DNA or chromatin, the method could find binding sites of transcription factors and/or chromatin-binding proteins that elude detection by ChIP-seq. Further, with an inducible M.EcoGII fusion protein, one could track processes over time like the spread of heterochromatin proteins during X-chromosome inactivation, the movement of cohesin and condensin complexes along chromosomes during chromosome pairing and condensation, and the homology search during homologous recombination. Improvements in Nanopore technology and modified base-calling software will likely extend the method's utility to other processes that have directional movement along chromosomes.

# Materials and methods

## Key resources table

| Reagent type (species) or resource | Designation | Source or reference | Identifiers | Additional information |
|---|---|---|---|---|
| Strain, strain background (*Saccharomyces cerevisiae*) | Various | This paper | NCBITaxon:4932 | W303 Background; see *Supplementary file 1* |
| Antibody | Anti-V5 (mouse monoclonal) | Thermo Fisher Scientific | Cat # R960-25 | (5 µl for ChIP, 1:5000 for Western) |
| Antibody | Anti-Hexokinase (rabbit polyclonal) | Rockland | Cat # 100-4159 | (1:20,000) |
| Antibody | Anti-m6A (rabbit polyclonal) | Synaptic Systems | Cat # 202-003 | 300 ng per 1 µg IP DNA |
| Antibody | Anti-Pgk1 (mouse monoclonal) | Invitrogen | Cat # 459,250 | (1:40,000) |
| Antibody | Anti-Sir3 (rabbit polyclonal) | N. Dhillon & R. Kamakaka | | (10 µl for ChIP, 1:1000 for Western) |
| Recombinant DNA reagent | SIR2-M.ECOGII | This paper | | Fusion of *SIR2* and bacterial adenine methyltransferase *M.ECOGII* |
| Recombinant DNA reagent | SIR3-M.ECOGII | This paper | | Fusion of *SIR3* and bacterial adenine methyltransferase *M.ECOGII* |
| Recombinant DNA reagent | SIR4-M.ECOGII | This paper | | Fusion of *SIR4* and bacterial adenine methyltransferase *M.ECOGII* |
| Recombinant DNA reagent | YEp24 | doi:https://doi.org/10.1016/0378-1119(79)90004-0 | | Empty 2 µm overexpression plasmid; see *Supplementary file 2* |
| Recombinant DNA reagent | pMB34 | This paper | | *M.ECOGII* inserted into pFA6a-natMX6; see *Supplementary file 2* |
| Recombinant DNA reagent | pMB54 | This paper | | *SIR3-M.ECOGII* inserted into YEp24; see *Supplementary file 2* |
| Recombinant DNA reagent | pFA6a-natMX6 | doi:10.1002/yea.1291 | | natMX6 marker integration cassette; see *Supplementary file 2* |
| Sequence-based reagent | Various oligonucleotides | This paper | qPCR and cloning primers | See *Supplementary file 3* |
| Commercial assay or kit | NEBNext Ultra II Library Prep Kit for Illumina | New England Biolabs | Cat # E7645 | |
| Commercial assay or kit | NEBNext Multiplex Oligos for Illumina | New England Biolabs | Cat # E7335/E7500 | |
| Commercial assay or kit | Qubit dsDNA HS | Invitrogen | Cat # Q32854 | |
| Commercial assay or kit | Qiaquick PCR Purification Kit | Qiagen | Cat # 28,104 | |
| Commercial assay or kit | Accel-NGS 1S Plus DNA Library Kit | Swift Biosciences | Cat # 10,024 | |
| Commercial assay or kit | Swift Single Indexing Primers Set A | Swift Biosciences | Cat # X6024 | |

*Continued on next page*

*Continued*

| Reagent type (species) or resource | Designation | Source or reference | Identifiers | Additional information |
|---|---|---|---|---|
| Commercial assay or kit | YeaStar Genomic DNA Kit | Genesee Scientific | Cat # 11-323 | |
| Commercial assay or kit | Covaris g-TUBE | Covaris | Cat # 520,079 | |
| Commercial assay or kit | NEB Oxford Nanopore Companion | New England Biolabs | Cat # E7180S | |
| Commercial assay or kit | NEBNext Quick Ligation Reaction Master Mix | New England Biolabs | Cat # B6058 | |
| Commercial assay or kit | NEB Blunt/TA Ligase Master Mix | New England Biolabs | Cat # M0367 | |
| Commercial assay or kit | Oxford Ligation Sequencing Kit | Oxford Nanopore Technologies | Cat # SQK-LSK109 | |
| Commercial assay or kit | Oxford Native Barcoding Expansion 1–12 | Oxford Nanopore Technologies | Cat # EXP-NBD104 | |
| Commercial assay or kit | Qiagen RNeasy kit | Qiagen | Cat # 74,104 | |
| Commercial assay or kit | SuperScript III First-Strand Synthesis System | Invitrogen | Cat # 18080051 | |
| Commercial assay or kit | DyNAmo HS SYBR Green kit | Thermo Fisher | Cat # F410L | |
| Commercial assay or kit | RNase-free DNase set | Qiagen | Cat # 79,254 | |
| Software, algorithm | SAMtools | doi:10.1093/bioinformatics/btp352 | | |
| Software, algorithm | Bowtie2 | doi:10.1038/nmeth.1923 | | |
| Software, algorithm | IGV | doi:10.1093/bib/bbs017 | | |
| Software, algorithm | Guppy Basecaller | Oxford Nanopore Technologies | | |
| Software, algorithm | Megalodon | Oxford Nanopore Technologies | | https://github.com/nanoporetech/megalodon |
| Software, algorithm | All-context rerio model | Oxford Nanopore Technologies | res_dna_r941_min_modbases-all-context_v001.cfg | For Megalodon modified base-calling; https://github.com/nanoporetech/rerio |
| Peptide, recombinant protein | Protein A Dynabeads | Thermo Fisher | Cat # 10,002D | |

*Continued on next page*

*Continued*

| Reagent type (species) or resource | Designation | Source or reference | Identifiers | Additional information |
|---|---|---|---|---|
| Peptide, recombinant protein | Proteinase K | New England Biolabs | Cat # P8107S | |
| Peptide, recombinant protein | RNase A | Thermo Fisher | Cat # EN0531 | |

## Strains

All strains in this study were derived from W303 (*Supplementary file 1*). For biological replicates, independent cultures were started from the same strain or isogenic strains as indicated by strain numbers in figure legends. *M.ECOGII* integrations were created by one-step integration of a PCR-amplified *M.ECOGII-natMX* cassette from pJR3525 using the primers listed in *Supplementary file 3*. Deletion of the BAH domain of *SIR3* was done using CRISPR-Cas9 gene editing using the guide RNA and repair template listed in *Supplementary file 3*. The guide RNA target and nontarget strands were integrated into a single guide RNA dropout-Cas9 expression plasmid (pJR3428, *Brothers and Rine, 2019*) by Golden Gate cloning, using the restriction enzyme *Bsm*BI as described in *Lee et al., 2015*. The repair template was made by annealing oligos described in *Supplementary file 3* and extending the 3′ ends using Phusion Polymerase (New England Biolabs, Beverly, MA). *SIR3* overexpression strain and its control strain were created by transformation and maintenance of 2 μm plasmids pJR3526 and YEp24, respectively.

## Plasmids

Plasmids used in this study are listed in *Supplementary file 2*. The *M.ECOGII-NatMX6* tagging plasmid (pJR3525) was made using standard Gibson cloning into pFA6a-natMX6 (*Hentges et al., 2005*). The codon-optimized *M.ECOGII* ORF with homology to the vector backbone was on a gene block (*Supplementary file 3*) from Integrated DNA Technologies (IDT, Coralville, IA) and was inserted into pFA6a-natMX6 linearized by PCR. The *SIR3p-SIR3-M.ECOGII* overexpression plasmid (pJR3526) was made using standard Gibson cloning into YEp24, a 2μ yeast expression plasmid carrying a *URA3* selectable marker (*Botstein et al., 1979*). The sequences for each plasmid are available in Supplementary Materials as fasta and.dna files.

## ChIP-seq

### Sample collection

Strains were grown to mid-log phase in YPD at 30°C. Approximately $10^9$ cells were collected, washed, and fixed for 15 min at 30°C in a final concentration of 1% formaldehyde. Fixation was quenched with a final concentration of 300 mM glycine for 10 min at 30°C. Cells were washed 1× with PBS and 2× with FA lysis buffer (50 mM HEPES, pH7.5, 150 mM NaCl, 1 mM EDTA, 1% [vol/vol] Triton X-100, 0.1% [wt/vol] sodium deoxycholate, 0.1% sodium dodecyl sulfate [SDS]) before flash freezing pellets in a 2-ml screw-cap tube. Pellets were resuspended in 1 ml FA lysis buffer and 500 μl of 0.5 mm zirconium ceramic beads (BioSpec Products, Bartlesville, OK, Cat # 11079105z) were added. Resuspended cells were bead beat with in a FastPrep-24 (MP Biomedicals, Burlingame, CA) at 5.5 amplitude, four cycles of 40 s ON/2 min on ice. Each tube was punctured at the bottom with a hot 20-G needle and placed into a new 1.5 ml tube, and sample was spun out of the tube into the new tube by spinning at $150 \times g$ for 1 min. The sample was moved into a 15-ml Bioruptor sonication conical tube with 100 μl of Bioruptor sonication beads (Diagenode, Denville, NJ, Cat # C01020031) and sonicated using the Bioruptor Pico (Diagenode) for 10 cycles of 30 s ON/30 s OFF.

### Immunoprecipitation

The sonicated extract was moved into a new 1.5 ml tube and spun at $16,000 \times g$ for 15 min at 4°C. The supernatant was moved into a new 1.5 ml tube and adjusted to 1 ml volume with FA lysis buffer. 50 μl of sample was taken aside as input, and then 25 μl of 20 mg/ml BSA and 5 μl of mouse anti-V5

antibody (Invitrogen, Waltham, MA, Cat # R960-25) or 10 µl of polyclonal rabbit anti-Sir3 antibody (from N. Dhillon & R. Kamakaka) was added to the rest of the sample and rotated overnight at 4°C. 50 µl of Protein A magnetic Dynabeads (Thermo Fisher, Waltham, MA, Cat # 10,002D) were added to the sample and rotated at 4°C for 1 hr. Magnetic beads were immobilized using a magnetic rack and washed by resuspension in 1 ml of various buffers in the following order: FA lysis buffer + 0.05% Tween-20, FA lysis buffer + 0.05% Tween-20 + 0.25 mM NaCl, ChIP wash buffer (10 mM Tris, pH 8.0, 0.25 M LiCl, 1 mM EDTA, 0.5% Nonidet P-40, 0.5% sodium deoxycholate, 0.05% Tween-20), and TE (10 mM Tris, pH 8.0, 1 mM EDTA) + 0.05% Tween-20. The washed beads were resuspended in 130 µl of ChIP elution buffer (10 mM Tris, pH 7.5, 1 mM EDTA, 1% SDS) and incubated at 65°C shaking at 900 rpm overnight. The next day, 2.5 µl of 10 mg/ml Proteinase K (New England Biolabs, Ipswich, MA, Cat # P8107S) and 2.5 µl of 10 mg/ml RNase A (Thermo Fisher, Cat # EN0531) were added to the elution and incubated at 42°C for 2 hr. Beads were immobilized on a magnetic rack and the supernatant containing the desired DNA to be sequenced was taken and purified using 1× (vol/vol) SPRI Select magnetic beads (Beckman Coulter, Brea, CA, Cat # B23317) according to the manufacturer's instructions.

## Library preparation and sequencing
Samples were prepared for sequencing using NEBNext Ultra II DNA Library Prep Kit for Illumina (New England Biolabs, Cat # E7645) according to the manufacturer's instructions. Samples were multiplexed using NEBNext Multiplex Oligos for Illumina (New England Biolabs, Cat # E7335/E7500). Library-prepped samples were sequenced on a MiniSeq System (Illumina, San Diego, CA).

## Analysis
Sequencing reads were aligned to the S288C sacCer3 reference genome (release R64-2-1_20150113, http://yeastgenome.org/), modified to include *matΔ* using Bowtie2 with the options '`--local --soft-clipped-unmapped-tlen --no-unal --no-mixed --no-discordant`' (*Langmead and Salzberg, 2012*). Reads were normalized to the genome-wide median, excluding rDNA, chromosome III, and subtelomeric regions (the first and last 10 kb of each chromosome). Analysis was performed using custom Python scripts modified from *Goodnight and Rine, 2020* (*Source code 1*, *Source code 2*) and displayed using IGV (*Thorvaldsdóttir et al., 2013*).

## **DIP-seq**
## DNA extraction
Cells were grown to mid-log phase in YPD, Complete Supplement Mixture (CSM), or CSM without Uracil (Sunrise Science Products, Knoxville, TN) at 30°C. Approximately $10^9$ cells were pelleted by centrifugation at 3200 × *g* for 2 min, washed with 1 ml of water, moved to a 2-ml screw-cap tube, and flash frozen. Cells were resuspended in 400 µl of Triton SDS Lysis Buffer (10 mM Tris, pH 8.0, 100 mM NaCl, 1 mM EDTA, 2% Triton X-100, 1% SDS), and 400 µl of phenol:chloroform:isoamyl alcohol 25:24:1 and 300 µl of 0.5 mm zirconium ceramic beads (BioSpec Products, Cat # 11079105z) were added to the resuspension. Cells were lysed by bead beating with in a FastPrep-24 (MP Biomedicals) at 5.5 amplitude, 4 cycles of 40 s ON/2 min on ice. The aqueous and organic phases were separated by centrifugation at 21,000 × *g* for 5 min, and the aqueous phase was moved to a new 1.5 ml tube. 400 µl of chloroform was added, vortexed at top speed for ~10 s, and spun down at 21,000 × *g* for 5 min to separate the aqueous and organic phases. The aqueous phase was moved to a new 1.5 ml tube, and 1 ml of 100% ethanol was added to precipitate nucleic acids. The sample was incubated at 4°C for 10–15 min and then spun down at 21,000 × *g* for 2 min to pellet the precipitated nucleic acids. Supernatant was discarded, the pellet was air-dried, and then the pellet was resuspended in 400 µl of TE (10 mM Tris–HCl, pH 8.0, 1 mM EDTA) + 4 µl of 10 mg/ml RNase A (Thermo Fisher, Cat # EN0531) and incubated at 37°C for 1 hr. 1 ml of 100% ethanol + 10 µl of 4 M ammonium acetate were added to the RNase solution and incubated at 4°C for 10–15 min to precipitate DNA. The precipitate was pelleted by centrifugation at 21,000 × *g* for 2 min, washed 1× with 70% ethanol, air-dried, and resuspended in 150–300 µl of water.

## Sonication

DNA concentration was measured using Qubit dsDNA HS reagents (Invitrogen, Cat #Q32854), and 6 µg of DNA was diluted to 20 ng/µl in 300 µl of water in 1.5 ml Bioruptor Pico Microtubes for sonication (Diagenode, Cat # C30010016). DNA was sonicated using a Bioruptor Pico (Diagenode) for 18 cycles of 15 s ON/90 s OFF. The sonicated DNA was moved to a new 1.5 ml tube.

## m$^6$A IP

DNA was denatured by incubating at 95°C for 10 min and then immediately placed on ice for 5 min. 200 µl of cold water and 500 µl of cold 5× DIP buffer (50 mM NaPO$_4$, pH 7.0, 700 mM NaCl, 0.25% Triton X-100) were added to bring the volume up to 1 ml. 50 µl was taken aside as input. 25 µl of 20 mg/ml BSA and 1.8 µg of antibody (Synaptic Systems rabbit anti-m$^6$A, Cat 202–003) were added to the rest of the sample and rotated overnight at 4°C. 50 µl of Protein A magnetic Dynabeads (Thermo Fisher, Cat # 10,002D) were added to the sample and rotated at 4°C for 1 hr. Magnetic beads were immobilized using a magnetic rack and washed by resuspension and rotation for 5 min at 4°C in 1 ml of various cold buffers in the following order: 2× with 1× DIP buffer + 0.05% Tween-20, 1× with 1× DIP buffer. For elution, beads were resuspended in 190 µl of DIP digestion buffer (50 mM Tris–HCl, pH 8.0, 10 mM EDTA, 0.5% SDS) + 10 µl of 10 mg/ml Proteinase K (New England Biolabs, Cat # P8107S). DIP digestion buffer was added to input samples up to 200 µl. Both the input and IP samples were incubated at 50°C for 2 hr and then cleaned up using the Qiaquick PCR Purification Kit (Qiagen, Hilden, Germany, Cat # 28104) and eluted in 35 µl of water.

## Library preparation and sequencing

Samples were prepared for sequencing using Accel-NGS 1S Plus DNA Library Kit (Swift Biosciences, Ann Arbor, MI, Cat # 10024) according to the manufacturer's instructions. Samples were multiplexed using Swift Single Indexing Primers Set A (Swift Biosciences, Cat # X6024). Library-prepped samples were sequenced on a MiniSeq System (Illumina).

## Analysis

Analysis was done as described in the section on ChIP-seq above.

# Nanopore sequencing

## DNA extraction

Cells were grown to mid-log phase in YPD, CSM (Sunrise Science Products), or CSM without Uracil at 30°C. Approximately 10$^8$ cells were pelleted, washed with 1 ml of water, and pellets were flash frozen. gDNA was extracted using the YeaStar Genomic DNA Kit (Genesee Scientific, San Diego, CA, Cat #11-323) according to the manufacturer's 'Protocol 1'. Specifically, thawed cell pellets were resuspended in 240 µl of YD digestion buffer + 10 µl R-Zymolyase and incubated at 30°C for 1 hr. 240 µl of YD Lysis buffer was added to the solution and vortexed at top speed for 15 s. 500 µl of chloroform was added to the solution and vortexed at top speed for 10 s and then inverted 10 times. The aqueous and organic phases were separated by centrifugation at 10,000 × *g* for 2 min, and the aqueous phase was equally separated into two ZymoSpin columns. ZymoSpin columns were spun at 10,000 × *g*, washed 2× with 300 µl of DNA wash buffer, and DNA was eluted from each column with 75 µl of water. Eluates were combined. DNA was sheared to ~15–20 kb by spinning through a Covaris g-TUBE (Covaris Inc, Woburn, MA, Cat # 520079) at 4200 rpm for 1 min, and repeating 1× with the tube flipped the other way in an Eppendorf Centrifuge 5424, according to the Covaris protocol. DNA was purified and concentrated using 1× (vol/vol) SPRI Select beads (Beckman Coulter, Cat # B23317) and eluted in 50 µl of water according to the manufacturer's instructions. DNA concentration was measured using Qubit dsDNA HS reagents (Invitrogen, Cat # Q32854).

## Library preparation and sequencing

Approximately 1–3 µg of purified, sheared genomic DNA was library prepped using the following reagents: NEB Oxford Nanopore Companion (New England Biolabs, Cat # E7180S), NEB Blunt/TA Ligase Master Mix (New England Biolabs, Cat # M0367), NEBNext Quick Ligation Reaction Master Mix (New England Biolabs, Cat # B6058), Oxford Ligation Sequencing Kit (Oxford Nanopore Technologies,

Oxford, UK, Cat # SQK-LSK109), and the Oxford Native Barcoding Expansion 1–12 (Oxford Nanopore Technologies, Cat # EXP-NBD104). The library was prepared and sequenced according to Oxford Nanopore's protocol for Ligation Sequencing Kit + Native Barcoding Expansion 1–12. Sequencing was done on a MinION sequencer with v9.4 flow cells (Oxford Nanopore Technolgies, Cat # FLO-MIN106).

### Analysis

Base-calling was first done using Guppy v5.0.11 using the high-accuracy model (dna_r9.4.1_450bps_hac.cfg), and reads were demultiplexed using guppy_barcoder. Read IDs corresponding to each barcode were extracted and written to a.txt file using a custom Python script (*Source code 3*). Reads corresponding to each barcode were aligned to the S288C reference genome (release R64-2-1_20150113, yeastgenome.org, modified to include *matΔ*) and modifications called with Megalodon (https://github.com/nanoporetech/megalodon, v2.3.3; *Stoiber, 2021a*) using the all-context rerio model (https://github.com/nanoporetech/rerio, res_dna_r941_min_modbases-all-context_v001.cfg; *Stoiber, 2021b*) and the flags `--mod-motif` 'Y A 0', `--files_out` 'basecalls mod_mappings per_read_mods', and `--read-ids-filename` 'barcodeXX_readIDs.txt' (the file that contained the extracted list of readIDs for a given barcode).

Results were aggregated into.bed files using 'megalodon_extras aggregate run', and these files were used for aggregate Nanopore plots. Before plotting, aggregated data were filtered to include only adenines with at least 10× coverage, and lines were smoothed using base R loess() function with enp.target = 100 and weighted by the coverage at each position.

Assessment of linker-region preference of Sir3–M.EcoGII used nucleosome-occupancy data from GEO Accession GSE97290 (*Chereji et al., 2018*).

The per-read database from Megalodon was converted into a.txt file using 'megalodon_extras per_read_text modified_bases'. For ease of use in RStudio, the data for each chromosome were extracted into its own.txt file using custom bash and awk scripts (*Source code 4*) and these files were used for single-read Nanopore plots. The probabilities output by Megalodon were made binary by calling adenines with a > 0.8 probability of being methylated as 'm6A' and all others 'A'.

The R scripts (as.html files) used to create each figure can be found in *Source code 5*.

### Limitations

This method showed possible limitations for some contexts: (1) The expression level of the fusion protein could increase the levels of background methylation. We found this to be true with the Sir2–M.EcoGII fusion protein, which is likely expressed at a higher level than Sir3–M.EcoGII due to the higher level of endogenous Sir2 expression. The level of methylation by Sir2–M.EcoGII in heterochromatin regions was higher than by Sir3–M.EcoGII, and background levels of methylation outside of heterochromatin regions were also higher than by Sir3–M.EcoGII. Importantly, the signal at heterochromatin was evident above even this raised background methylation. (2) Methylation at the level of single reads was variable and spotty, possibly due to at least two contributors. There may be occupancies that are too transient to allow methylation. Secondly, computational limitations for calling modified adenines without a guiding sequence motif meant that lower-confidence (probably <0.8) m⁶A calls were not considered methylated. At present, qualitative conclusions based on single-read data can be made with confidence, but as Nanopore technology improves, single-read data will become more amenable to statistical and spatial analysis. (3) Because this method can capture transient interactions better than methods like ChIP-seq it may overestimate degrees of occupancy unless combined with DIP- or ChIP-seq.

## Reverse Transcription and Quantitative PCR (RT-qPCR)

### RNA extraction

Cells were grown to mid-log phase in YPD, CSM (Sunrise Science Products), or CSM–Uracil at 30°C, and RNA was extracted using the Qiagen RNeasy kit (Qiagen, Cat # 74104) according to the manufacturer's instructions for purification of total RNA from yeast. Briefly, ~6 × 10⁷ cells were resuspended in 600 μl of buffer RLT, 500 μl of 0.5 mm zirconium ceramic beads (BioSpec Products, Cat # 11079105z) were added, and cells were lysed by bead beating with in a FastPrep-24 (MP Biomedicals) at 5.5 amplitude, 3 cycles of 40 s ON/2 min on ice. Cells were pelleted by spinning at 21,000 × *g* for 2 min, and the supernatant was moved to a new tube. One volume of 70% ethanol was added to the

supernatant and the sample was spun through an RNeasy spin column. The column was washed with 350 µl of buffer RW1, then 10 µl of DNase + 70 µl of buffer RDD (Qiagen, Cat # 79256) were added to the column and incubated for 15 min at room temperature. 500 µl of buffer RW1 was added and spun through the column. The column was then washed with 500 µl of buffer RPE 2×, and RNA was eluted with 80–150 µl of RNase-free water.

## RT-qPCR

Complementary DNA was synthesized using the SuperScript III First-Strand Synthesis System (Invitrogen, Cat # 18080051) and oligo(dT) primers according to the manufacturer's protocols. Quantitative PCR of complementary DNA was performed using the DyNAmo HS SYBR Green kit (Thermo Fisher, Cat # F410L) on an Mx3000P machine (Stratagene, La Jolla, CA) using the primers listed in **Supplementary file 3**. Standard curves were generated using a tenfold dilution series of one of the prepared samples.

## Protein immunoblotting

Each strain was grown to saturation overnight in 5 ml YPD. Overnight cultures were diluted to ~2 × $10^5$ cells/ml in fresh YPD, grown to mid-log phase, and ~$10^8$ cells were harvested and pelleted. Pellets were resuspended in 1 ml of 5% trichloroacetic acid and incubated at 4°C for 10–30 min. The precipitates were pelleted, washed once with 1 ml of 100% acetone, and air-dried. Dried pellets were resuspended in 100 µl of protein breakage buffer (50 mM Tris–HCl, pH 7.5, 1 mM EDTA, 3 mM DTT) and an equal volume of 0.5 mm zirconium ceramic beads (BioSpec Products, Cat # 11079105z) followed by four cycles of 40 s bead beating/2 min on ice in a FastPrep-24 (MP Biomedicals). 100 µl of 2× Laemmli buffer (120 mM Tris–HCl, pH 7.5, 20% glycerol, 4% SDS, 0.02% bromophenol blue, 10% beta-mercaptoethanol) was added to each sample and incubated at 95°C for 5 min. Insoluble material was pelleted by centrifugation and an equal volume of the soluble fraction from each sample was run on an SDS–polyacrylamide gel (Mini-PROTEAN TGS Any kD precast gel; BioRad, Hercules, CA Cat # 4569033) and transferred to a nitrocellulose membrane using a TransBlot Turbo Mini 0.2 µm Nitrocellulose Transfer Pack (BioRad, Cat # 1704158) on the High MW setting of a TransBlot Turbo machine (BioRad). The membrane was blocked in Intercept Blocking Buffer (LI-COR Biosciences, Lincoln, NE, Cat # 927-70001), and the following primary antibodies and dilutions were used for detection: V5 (R960-25, 1:5000; Invitrogen), Hxk2 (#100-4159, 1:20,000; Rockland Immunochemicals Inc, Pottstown, PA), Sir3 (1:1000; rabbit polyclonal from N. Dhillon & R. Kamakaka), and Pgk1 (Cat # 459250, 1:40,000; Invitrogen). The secondary antibodies used were IRDye 800CW (926-32210) and 680RD (926-68071) (1:20,000; LI-COR Biosciences), and the membrane was imaged on a LI-COR Odyssey Imager. All washing steps were performed with PBS + 0.1% Tween-20.

## Acknowledgements

We would like to thank the Rine Lab for lively discussions and helpful comments through the development and execution of this project, especially Davis Goodnight. We also especially thank Marc Fouet for generous technical assistance with lab servers. We thank Namrita Dhillon and Rohinton Kamakaka for graciously sending us Sir3 antibody. We thank Koen Van den Berge and Sandrine Dudoit for advice on statistics and new ways of thinking about and approaching our Nanopore data. We thank Marcus Stoiber at Oxford Nanopore for help and advice with Megalodon. This research used the Savio computational cluster resource provided by the Berkeley Research Computing program at the University of California, Berkeley (supported by the UC Berkeley Chancellor, Vice Chancellor for Research, and Chief Information Officer). This work was funded by grants from the National Institutes of Health to JR (R35 GM139488). MB received support from a National Science Foundation Graduate Research Fellowship (Grant No. 1752814).

## Additional information

### Funding

| Funder | Grant reference number | Author |
|---|---|---|
| National Science Foundation | 1752814 | Molly Brothers |
| National Institutes of Health | R35 GM139488 | Jasper Rine |

The funders had no role in study design, data collection, and interpretation, or the decision to submit the work for publication.

### Author contributions

Molly Brothers, Conceptualization, Formal analysis, Funding acquisition, Investigation, Methodology, Visualization, Writing - original draft, Writing - review and editing; Jasper Rine, Conceptualization, Funding acquisition, Resources, Supervision, Writing - review and editing

### Author ORCIDs

Molly Brothers http://orcid.org/0000-0002-6114-3393
Jasper Rine http://orcid.org/0000-0003-2297-9814

### Decision letter and Author response

Decision letter https://doi.org/10.7554/eLife.75653.sa1
Author response https://doi.org/10.7554/eLife.75653.sa2

## Additional files

### Supplementary files

- Transparent reporting form
- Supplementary file 1. Strains used in this study.
- Supplementary file 2. Plasmids used in this study.
- Supplementary file 3. Oligonucleotides used in this study.
- Source code 1. Creation of bedgraph files from fastq files.
- Source code 2. Normalization of bedgraph files to genome-wide median.
- Source code 3. Extract nanopore read IDs corresponding to each barcode from sequencing summary file.
- Source code 4. Extract single read data corresponding to a given chromosome.
- Source code 5. R scripts used to construct nanopore data into figures.

### Data availability

All strains (Supplementary File 1) and plasmids (Supplementary File 2) are available upon request. Sequencing data is available in GEO under the SuperSeries GSE190137. ChIP-seq and DIP-seq data are under accession code GSE189038 in the SuperSeries. Nanopore data are under accession code GSE190136 in the SuperSeries.

The following dataset was generated:

| Author(s) | Year | Dataset title | Dataset URL | Database and Identifier |
|---|---|---|---|---|
| Brothers M, Rine J | 2021 | Distinguishing between recruitment and spread of silent chromatin structures in Saccharomyces cerevisiae | https://www.ncbi.nlm.nih.gov/geo/query.acc.cgi?acc=GSE190137 | NCBI Gene Expression Omnibus, GSE190137 |

The following previously published datasets were used:

| Author(s) | Year | Dataset title | Dataset URL | Database and Identifier |
|---|---|---|---|---|
| Brothers M, Rine J | 2021 | Distinguishing between recruitment and spread of silent chromatin structures in Saccharomyces cerevisiae [I] | https://www.ncbi.nlm.nih.gov/geo/query/acc.cgi?acc=GSE189038 | NCBI Gene Expression Omnibus, GSE189038 |
| Brothers M, Rine J | 2022 | Distinguishing between recruitment and spread of silent chromatin structures in Saccharomyces cerevisiae [II] | https://www.ncbi.nlm.nih.gov/geo/query/acc.cgi?acc=GSE190136 | NCBI Gene Expression Omnibus, GSE190136 |
| Chereji RV, Ramachandran S, Bryson TD, Henikoff S | 2017 | Precise genome-wide mapping of single nucleosomes and linkers in vivo | https://www.ncbi.nlm.nih.gov/geo/query/acc.cgi?acc=GSE97290 | NCBI Gene Expression Omnibus, GSE97290 |

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
