## [Editor Report]

This manuscript studies the mechanism of transcriptional silencing in *S. cerevisiae*. Using two new tools for this field, fusion of a silencing protein to a DNA methyltransferase and long-read Nanopore sequencing, the results have provided both technical advances and new insights into the role of Sir3 in this process.

---

## [Decision Letter]

**Decision letter after peer review:**

Thank you for submitting your article "Distinguishing between recruitment and spread of silent chromatin structures in *Saccharomyces cerevisiae*" for consideration by *eLife*. Your article has been reviewed by 3 peer reviewers, one of whom is a member of our Board of Reviewing Editors, and the evaluation has been overseen by Jessica Tyler as the Senior Editor. The reviewers have opted to remain anonymous.

Essential revisions:

All reviewers have found this manuscript interesting and the data of high quality. However, several concerns are raised and discussed among reviewers. All reviewers agree that the following two control experiments are essential for the revision.

1) The overexpression of Sir3 should be confirmed using western blot analysis.

2) Figure 3 – In the overexpression experiments, the authors are making conclusions about heterochromatin formation based on the level of methylation. It seems important to look at the different classes of telomeres by more than one method in order to fortify and expand this conclusion. First, looking at the level of recruitment of the Sir3-M.EcoGII by ChIP would tell us if the difference in methylation is caused at the level of recruitment. This is a Class A experiment as either result is interesting – either there's differential recruitment of Sir3 between chromosome 6 and the others or there's differential ability of Sir3 to spread/methylate. Second, if Sir3-M.EcoGII is recruited at a higher level, it would be worthwhile to look at the possible silencing of genes near the telomeres to see if their expression is reduced. This is especially important given the results presented in Figure 4 where a small change in association appears to increase silencing.

Please also address or respond to all the other concerns and suggestions made by reviewers in your revised manuscript and in your response to reviews.

*Reviewer #1:*

In this manuscript titled "Distinguishing between recruitment and spread of silent chromatin structures in *Saccharomyces cerevisiae*", the authors developed a highly sensitive technique using a non-sequence specific bacterial DNA methyltransferase fused with yeast silencing protein Sir3 to investigate the details of Sir3 binding and spreading at yeast silencing loci. They found that a truncated and silencing deficient Sir3 lacking the nucleosome binding BAH domain was still recruited to silencers/silencing nucleation sites but was unable to spread. Overexpressing Sir3 did not lead to further spreading at HML and HMR but had variable effects at telomeres. Finally, they provided evidence that transcriptional repression was established before Sir3 spreading.

A key strength in this study is the use of Sir3-M.EcoGII to map Sir3's interaction and spreading at silencing loci. This technique afforded high resolution and high sensitivity in detecting Sir3 binding and spreading than previously used approaches. When combined with BAH domain truncation, and inducible expression, the investigators were able to answer many questions on the mechanistic details. Hence, this study provides novel insights into Sir3-mediated transcription silencing and transcription silencing mechanism in general.

The data presented in this manuscript are generally convincing and support the main conclusions.

1. As the authors mentioned, the Sir3 BAH domain deletion results in silencing defects and lack of Sir3 spreading. Importantly, the study reports that sir3-bah∆ can still bind to silencers/silencing nucleation sites. So it appears that transcription silencing is associated with Sir3 spreading. Interestingly, in the Sir3 temperature sensitive experiment, the authors showed that transcriptional repression is already established prior to Sir3 spreading. How would the authors interpret the relationship between Sir3 spreading and transcription silencing? What are the roles of the BAH domain in transcriptional repression beyond Sir3 spreading?

2. The authors described several identified functional domains in Sir3. In addition to the BAH domain, there are AAA+ and wH domains on Sir3. However, only BAH domain deletion was investigated. Could the functions of the other domains be investigated using this Sir3- M.EcoGII approach? The AAA+ domain interacts with Sir4 and the wH domain mediates dimerization, hence both are relevant in the investigation of Sir3 mediated silencing and spreading.

*Reviewer #2:*

Heterochromatin is a repressive chromatin structure that is first nucleated and then spreads across large domains to enforce gene silencing. Heterochromatin coats repetitive elements as well as gene-rich regions to establish the gene silencing pattern that maintains cell identity. In this manuscript, the authors investigate the coordinated relationship between initial nucleation and spreading activity and the first repressive effects elicited by the SIR complex in budding yeast, where it represses transcription of genes at HML, HMR and telomeres. The authors create a fusion protein consisting of Sir3, which is the major factor driving heterochromatin spread and silencing, and the bacterial methyladenosine methyltransferase. Expression of this protein combined with genome-wide sequencing techniques allowed the authors to map Sir3 chromatin interactions. By deleting the Sir3 nucleosome binding domain on the fusion protein, or by creating a temperature-sensitive mutant, the authors probe the functional mechanisms underlying spreading/silencing. They find that spreading requires Sir3 nucleosome binding, not simply recruitment, and that overexpression of Sir3 shows only a variable and limited effect on spreading at different sites – arguing against a simple dose-dependency. Finally, they find that gene repression can be detected prior to full occupancy/spread of Sir3 across domains.

Critique: Overall, work described in this manuscript is interesting. Results presented are of high quality and support the main conclusions.

I recommend publication of this exciting work in *eLife*.

*Reviewer #3 (Recommendations for the authors):*

In this manuscript, the authors study silencing in *S. cerevisiae* using a novel approach. They construct a functional fusion of Sir3 to the adenine methyltransferase M.EcoGII, resulting in DNA methylation as a mark of Sir3 association with chromatin. Then, they use Nanopore long-read sequencing to obtain high-resolution analysis of this recruitment. Using these tools they provide several new and interesting results. By comparing Sir3-M.EcoGII ChIP to methylation, they provide evidence that methylation is a more sensitive assay, as it is detectable across silenced regions. Then, using a Sir3 mutant, lacking its BAH domain, they show that this mutant is capable of being recruited, but not of spreading. Finally, they provide evidence that overexpression of Sir3 does not increase the extent of telomere-associated heterochromatin domains for most telomeres, a result that contrasts with much older studies, thus providing a new view of the consequences of increased Sir3 levels.

1. lines 114-117 and Figure 1B – It's not clear why silencing was assayed by fluorescence here, when it was assayed by RT-qPCR everywhere else. If there is an advantage, please provide it. Otherwise, it would be better to include RT-qPCR here as well to provide a basis for comparison with the later experiments. Also, some readers will not be familiar with using fluorescent reporters to assay silencing so if this stays, the authors should cite Dodson and Rine and give an estimate of the sensitivity of this assay compared to the older methods of measuring RNA levels or mating.

2. line 157, Figure 1D, E – The authors discuss a periodicity of reads as shown in Figure 1E. However, there is also a periodicity evident in Figure 1D. Do the peaks from the two methods match up? Please address this in this section.

3. Figure 3 – In the overexpression experiments, the authors are making conclusions about heterochromatin formation based on the level of methylation. It seems important to look at the different classes of telomeres by more than one method in order to fortify and expand this conclusion. First, looking at the level of recruitment of the Sir3-M.EcoGII by ChIP would tell us if the difference in methylation is caused at the level of recruitment. This is a Class A experiment as either result is interesting – either there's differential recruitment of Sir3 between chromosome 6 and the others or there's differential ability of Sir3 to spread/methylate. Second, if Sir3-M.EcoGII is recruited at a higher level, it would be worthwhile to look at the possible silencing of genes near the telomeres to see if their expression is reduced. This is especially important given the results presented in Figure 4 where a small change in association appears to increase silencing.

4. lines 282-283, Figure 4B – The authors wrote that, after the 37 to 25 shift, the Sir3-8 protein levels grew "to match" the constitutive 25C levels. However, their results show the level at 150 minutes to be 0.63 of the constitutive level. I suggest revising and simply giving the real value in the text, something like, "Over the course of 150 minutes, the protein levels of sir3-8 slowly increased to approximately 63% that of the level in constitutive 25C growth conditions (Figure 4B)."

---

## [Author Response]

Essential revisions:All reviewers have found this manuscript interesting and the data of high quality. However, several concerns are raised and discussed among reviewers. All reviewers agree that the following two control experiments are essential for the revision.1) The overexpression of Sir3 should be confirmed using western blot analysis.

We have added a Western blot for the Sir3-M.EcoGII overexpression experiments as a new Figure 3B (and added the source data with full uncropped blots as Figure 3–Source Data 1) along with the following text in the results:

“We confirmed the overexpression of *SIR3-M.ECOGII* in these strains by protein immunoblotting (Figure 3B)”

We have also updated the Materials and Methods to add the new antibody we used. (Thanks to Namrita Dhillon and Rohinton Kamakaka for the Sir3 antibody we used for the new Western and ChIP-seq experiments!)

2) Figure 3 – In the overexpression experiments, the authors are making conclusions about heterochromatin formation based on the level of methylation. It seems important to look at the different classes of telomeres by more than one method in order to fortify and expand this conclusion. First, looking at the level of recruitment of the Sir3-M.EcoGII by ChIP would tell us if the difference in methylation is caused at the level of recruitment. This is a Class A experiment as either result is interesting – either there's differential recruitment of Sir3 between chromosome 6 and the others or there's differential ability of Sir3 to spread/methylate. Second, if Sir3-M.EcoGII is recruited at a higher level, it would be worthwhile to look at the possible silencing of genes near the telomeres to see if their expression is reduced. This is especially important given the results presented in Figure 4 where a small change in association appears to increase silencing.

We have followed the reviewers’ suggestion and backed up the methylation data upon overexpression of Sir3-EcoGII with ChIP data on the binding of Sir3-EcoGII. We feel confident concluding that the ChIP-seq results at *HML*, *HMR*, and telomeres parallel the Nanopore data quite well, suggesting that the increase in methylation signal by Nanopore sequencing at telomere 6R is due to Sir3 binding/contacting chromatin more frequently.

We have included ChIP-seq results for the overexpression experiments in a new Figure 3—figure supplement 1 and added the following text to the Results section:

“The changes in methylation upon overexpression of Sir3-M.EcoGII matched changes in occupancy of the fusion protein by ChIP-seq, suggesting that methylation reflects binding of the protein well (Figure 3—figure supplement 1).”

“As at *HML* and *HMR,* the changes in methylation seen with Nanopore sequencing matched with changes in occupancy measured by ChIP-seq (Figure 3—figure supplement 1).”

We have also updated the Materials and Methods to add the new antibody we used and submitted the new ChIP-seq data to the GEO to add to accession 189038.

The reviewer invited us to look at the impact of greater Sir3 spreading on gene expression. It turns out that nearly the entire body of knowledge on the impact of Sir3 overexpression on repression of telomeric genes comes from telomere 6R, which is one of those our work here established the physical basis of. In earlier work from the lab mapping Sir protein distribution and silencing on all yeast telomeres (Ellahi *et al.,* 2015), we established that there are few subtelomeric genes susceptible to silencing, and most of these are in gene families, making it difficult to attribute the modest effects observed to any one copy of that gene.

Please also address or respond to all the other concerns and suggestions made by reviewers in your revised manuscript and in your response to reviews.Reviewer #1 (Recommendations for the authors):1. As the authors mentioned, the Sir3 BAH domain deletion results in silencing defects and lack of Sir3 spreading. Importantly, the study reports that sir3-bah∆ can still bind to silencers/silencing nucleation sites. So it appears that transcription silencing is associated with Sir3 spreading. Interestingly, in the Sir3 temperature sensitive experiment, the authors showed that transcriptional repression is already established prior to Sir3 spreading. How would the authors interpret the relationship between Sir3 spreading and transcription silencing? What are the roles of the BAH domain in transcriptional repression beyond Sir3 spreading?

Indeed, *sir3-bah∆* resulted in silencing defects at both *HML* and *HMR*, but there was still a small amount of repression at *HML* and an approximately 10-fold repression at *HMR* compared to *sir3∆* (Figure 2C). In the Discussion section, we discussed the results of the *bah∆* and time course experiments and concluded that some repression is possible without Sir3 spreading, but spreading is required for full transcriptional silencing.

2. The authors described several identified functional domains in Sir3. In addition to the BAH domain, there are AAA+ and wH domains on Sir3. However, only BAH domain deletion was investigated. Could the functions of the other domains be investigated using this Sir3- M.EcoGII approach? The AAA+ domain interacts with Sir4 and the wH domain mediates dimerization, hence both are relevant in the investigation of Sir3 mediated silencing and spreading.

Great point! We had the same thought during this project and attempted to make and test various mutants in the BAH, wH, and AAA+ domains of Sir3: D17G (in BAH, no/little interaction with nucleosomes), K657A/K658A (in AAA+, no interaction with Sir4), and wH∆ (no dimerization). Unfortunately, these approaches failed for various reasons. In particular, the D17G and wH∆ mutants we made resulted in lower or completely absent Sir3 protein levels (by Western, Author response image 1). Previous work using these mutants showed wild-type levels of Sir3 (wH∆: Oppikofer *et al.,* 2013, D17G: Buchberger *et al.,* 2008, K657A/K658A: Ehrentraut *et al.,* 2011). We are unsure why our results are not consistent with these findings. The above mutant alleles were expressed from single-copy plasmids, whereas ours were expressed from the endogenous *SIR3* locus, so perhaps therein lies the difference.

**Author response image 1. sa2fig1:** 

Reviewer #3:

1. lines 114-117 and Figure 1B – It's not clear why silencing was assayed by fluorescence here, when it was assayed by RT-qPCR everywhere else. If there is an advantage, please provide it. Otherwise, it would be better to include RT-qPCR here as well to provide a basis for comparison with the later experiments. Also, some readers will not be familiar with using fluorescent reporters to assay silencing so if this stays, the authors should cite Dodson and Rine and give an estimate of the sensitivity of this assay compared to the older methods of measuring RNA levels or mating.

We have replaced the fluorescent assay results in Figure 1B with RT-qPCR results for each strain. The take-home message is the same, but the reviewer makes the point that the RT-qPCR data are probably more accessible to a wider audience.

2. line 157, Figure 1D, E – The authors discuss a periodicity of reads as shown in Figure 1E. However, there is also a periodicity evident in Figure 1D. Do the peaks from the two methods match up? Please address this in this section.

Thank you for pointing this out. You are correct that there appears to be a periodicity in Figure 1D, but the peaks in 1D are not the same ‘peaks’ as in 1E. Otherwise, there would be many more peaks in 1D. The data visualization in 1D (and all of the other aggregate nanopore data figures) is achieved using by applying a loess function, and thus the apparent periodicity in Figure 1D (and others) is likely a result of the same phenomenon seen in the single read data. If we zoom into the aggregate data and increase the granularity of the loess smoothing, more peaks and valleys appear in the aggregate data that match up with linker regions (Figure 1—figure supplement 3A).

3. Figure 3 – In the overexpression experiments, the authors are making conclusions about heterochromatin formation based on the level of methylation. It seems important to look at the different classes of telomeres by more than one method in order to fortify and expand this conclusion. First, looking at the level of recruitment of the Sir3-M.EcoGII by ChIP would tell us if the difference in methylation is caused at the level of recruitment. This is a Class A experiment as either result is interesting – either there's differential recruitment of Sir3 between chromosome 6 and the others or there's differential ability of Sir3 to spread/methylate. Second, if Sir3-M.EcoGII is recruited at a higher level, it would be worthwhile to look at the possible silencing of genes near the telomeres to see if their expression is reduced. This is especially important given the results presented in Figure 4 where a small change in association appears to increase silencing.

This point was addressed above in the “Essential Revisions” section. We reiterate the resulting changes here:

“The changes in methylation upon overexpression of Sir3-M.EcoGII matched changes in occupancy of the fusion protein by ChIP-seq, suggesting that methylation reflects binding of the protein well (Figure 3—figure supplement 1).”

“As at *HML* and *HMR,* the changes in methylation seen with Nanopore sequencing matched with changes in occupancy measured by ChIP-seq (Figure 3—figure supplement 1).”

4. lines 282-283, Figure 4B – The authors wrote that, after the 37 to 25 shift, the Sir3-8 protein levels grew "to match" the constitutive 25C levels. However, their results show the level at 150 minutes to be 0.63 of the constitutive level. I suggest revising and simply giving the real value in the text, something like, "Over the course of 150 minutes, the protein levels of sir3-8 slowly increased to approximately 63% that of the level in constitutive 25C growth conditions (Figure 4B)."

We modified the text to include this more specific statement.